



# Large interannual changes in supraglacial drainage basin areas and channels that flow downstream uphill: lessons from field surveys of moulin-connected streams on the Greenland Ice Sheet

Jessica Mejia[1], Jason Gulley[2], Celia Trunz[3], Charles Breithaupt[2], and Matthew Covington[4]

[1]Department of Earth and Environmental Sciences, Syracuse University, Syracuse, NY, USA
[2]Department of Geology, University of South Florida, Tampa, FL, USA
[3]Center for Hydrogeology, University of Neuchâtel, Neuchâtel, CH
[4]Department of Geosciences, University of Arkansas, Fayetteville, AR, USA

**Correspondence:** Jessica Mejia (jzmejia@syr.edu)

**Abstract.** Internally drained catchments (IDCs) define the ice surface area draining into a moulin. IDCs are thought to be controlled by the influence of basal topography on the ice surface, which should produce IDCs with static, topographically-defined catchment areas. Our observations of lakes overtopping drainage divides, fluvial incision through those drainage divides and connection of previously isolated adjacent lake basins suggests that supraglacial drainage basins are more complicated.

Here, we document interannual variability in the size, shape and density of IDCs in a 31.7 km$^2$ area by mapping supraglacial streams within three mid-elevation catchments on the Paakitsoq Region of the Greenland Ice Sheet in 2017 and 2018. In two of the three catchments, snow-infill of the previous year's incised streams diverted meltwater flow away from relic moulins, which rerouted flow over topographic divides and created new incised channels that flowed downstream against the surface topographic gradient and drained to different moulins than in the previous year. Catchment consolidation resulted the growth

of our central catchment from 8.2 km$^2$ in 2017, to 27.8 km$^2$ in 2018, and 31.7 km$^2$ in 2019, an area increase of 387% that was coincident with a decrease in the number of catchments, and moulins, decreasing from 3 to 1 within this area. Our results highlight that wintertime snowplug formation in supraglacial channels can change catchment-scale supraglacial hydrology and potentially impact hydrodynamic coupling across large areas of the ice sheet by turning moulins on and off.

## 1   Introduction

Mass loss from the Greenland Ice Sheet (GrIS) has been accelerating since the 1990's in response to climatic warming, and the GrIS has become one of the largest contributors to global sea level rise. The GrIS loses mass through meltwater runoff and dynamically through iceberg calving. While often treated separately, melting and ice dynamics are linked because moulins allow meltwater to reach the base of the ice sheet where it can alter subglacial water pressures and modulate sliding (Sundal et al., 2011). Although the supraglacial hydrological system regulates the timing and magnitude of meltwater inputs to the bed

(McGrath et al., 2011; Mejia et al., 2022; Smith et al., 2015; Willis et al., 2002; Yang et al., 2018), relatively little attention has been directed towards understanding the processes that govern the supraglacial drainage system.





Most of the meltwater produced on the GrIS surface collects within complex networks of supraglacial streams and rivers that transport meltwater downslope before terminating in moulins (Smith et al., 2015; Yang et al., 2016). Each stream network terminating into a single moulin is termed an internally drained catchment (IDC) (Yang and Smith, 2016), the area and shapes of which control the timing and volume of meltwater delivered to moulins (Banwell et al., 2013, 2016; McGrath et al., 2011; Mejia et al., 2022; Smith et al., 2015; Willis et al., 2002; Yang et al., 2018). Catchment geometry is largely dependent upon ice surface topography (Crozier et al., 2018; Karlstrom and Yang, 2016; Leeson et al., 2012), which, over large length-scales (~1–10 km), is controlled by the transfer of bed topography to the ice surface (Gudmundsson, 2003; Ignéczi et al., 2018; Lampkin, 2011; Raymond et al., 1995). Depressions at the bed roughly correspond to surficial depressions that form individual drainage basins that maintain their positions rather than advecting downglacier with ice flow (Echelmeyer et al., 1991; Morriss et al., 2013). In the absence of crevasses or moulins, meltwater will collect in surface depressions to form supraglacial lakes. These lakes can drain either quickly through new crevasses that intersect the lake (Mejía et al., 2021; Morriss et al., 2013) or slowly by flowing into neighboring catchments. Slow supraglacial lake drainage occurs when the lake water level surpasses the lowest elevation point of the drainage divide surrounding the lake and initiates a spillover event wherein lake water will thermally incise a channel though the drainage divide (Tedesco et al., 2013). Notably, thermally incised channels have been documented cutting through topographic divides separating individual drainage basins in Greenland (Smith et al., 2015; Yang et al., 2015) and on alpine glaciers (Gulley et al., 2009). Ultimately, channels incised through drainage divides allow water to enter into neighboring drainage basins where it will continue to flow downslope until reaching a crevasse or a moulin.

Moulin locations are typically regarded as static, either reactivating annually as ice advects downglacier (Andrews, 2015; Catania and Neumann, 2010) or forming within new crevasses that open in the same areas of high extensional strain. The pronounced influence of static basal topography on moulin locations, ice surface topography, and drainage network architecture underlies the widely held assumption that IDCs are spatially fixed from year to year (Yang et al., 2016). While the fluvial incision of meltwater channels can alter ice surface topography over short distances on the order of a few hundred meters, fluvial incision is not thought to impact catchment-scale flow routing because of the overwhelming adherence of stream flow to larger-scale ice surface topography (Crozier et al., 2018; Karlstrom and Yang, 2016).

Here we report on the coalescence of three neighboring catchments located in the Paakitsoq region of west Greenland between 2017 and 2019. We document interannual variability in the flow of the largest supraglacial streams and in terminal moulin location within each catchment by conducting roving differential-GPS (dGPS) surveys complemented with satellite remote sensing products. We find that snowplug formation within relic incised channels can divert flow away from previously active moulins, causing meltwater to collect within lakes or local depressions until it overflowed a drainage divide and connected the two previously isolated drainage basins through fluvial incision. Despite the short distances over which thermally incised channels directed flow in opposition to large-scale ice topography as they cut through drainage divides, overflow and incision was responsible for the coalescence of three previously isolated catchments over three years.





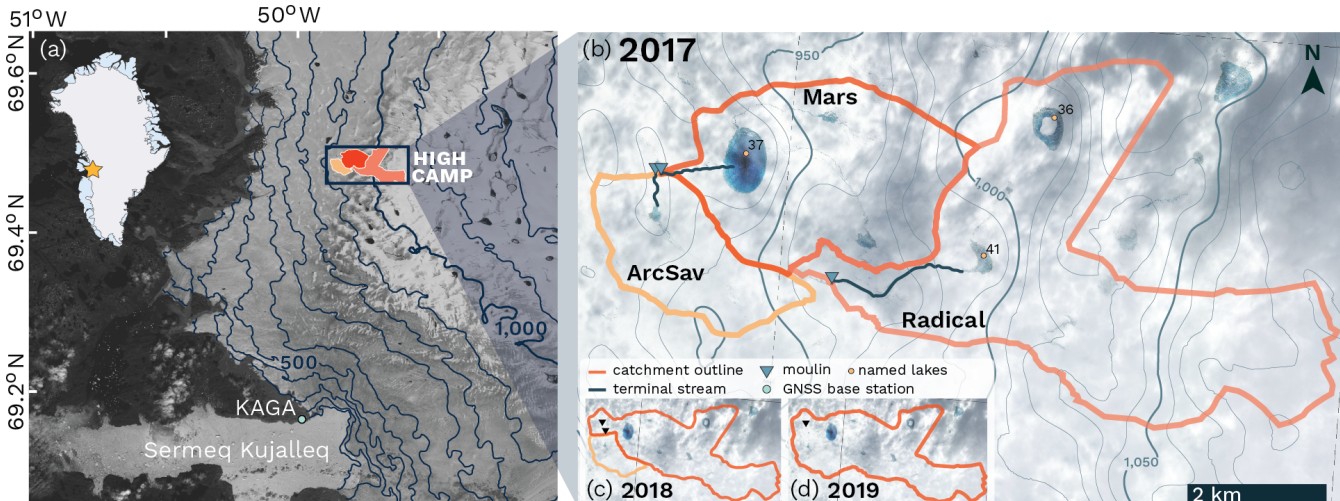

**Figure 1. High Camp field site, Paakitsoq region of the western Greenland Ice Sheet**. (a) Landsat-8 imagery courtesy of the US Geological Survey, acquired 21 July 2018, showing the location of our field site High Camp within Sermeq Avannarleq. Surface elevation contours derived from ArcticDEM in m above the WGS84 ellipsoid. (b–d) WorldView imagery 03 July 2017 (©2017 Maxar) overlaid with 2017 catchment boundaries. The highest-order stream for each catchment are traced in blue with and terminal moulins marked with triangles. Morriss et al. (2013) Named lakes (dots). Catchment outlines for (c) 2018 and (d) 2019 with terminal moulin locations (triangles).

## 2   Study Area

During the 2017 and 2018 melt seasons, we established a field camp adjacent to three mid-elevation (920–1,090 m) catchments in the ablation area of Sermeq Avannarleq in western Greenland, ~11 km from the ice margin (Fig. 1). The three catchments occupied a total area of 32 $km^2$. Each catchment contained a supraglacial lake known to drain slowly into moulins located outside of the lake basin (Banwell et al., 2012; Covington et al., 2020; Mejia et al., 2022; Morriss et al., 2013).

The smallest catchment, named ArcSav, drained an area of 3.9 $km^2$ in 2017 and 2018 (Fig. 1, Table 1). ArcSav Catchment

contained a single supraglacial lake, *ArcSav Lake*, located on the northern edge of the catchment at an elevation of ~900 m above the WGS84 ellipsoid (69.5498°N, -49.7682 °E). ArcSav Lake drained and refilled annually between 2017–2019 filling to a maximum area of 0.41 $km^2$ in 2018 and draining into moulins to the north.

The larger Mars Catchment was located on the eastern edge of ArcSav and drained an area of 8.2 $km^2$ in 2017 (Fig. 1). Mars Catchment contained a perennial supraglacial lake, *Mars Lake*, at an elevation of 914–921 m which has reoccupied the

same location since at least 2002 (Lake 37 in Morriss et al., 2013). In 2017–2019, Mars Lake drained through canyonized supraglacial streams originating at the western shoreline and terminating in moulins located outside of the basin containing Mars Lake.





| | Area (km$^2$) | | | Coordinates | | Elevation |
| | 2017 | 2018 | 2019 | °N | °W | m |
|---|---|---|---|---|---|---|
| ArcSav | 3.9 | 3.9 | ⟲ | 69.545 | 49.755 | 920–960 |
| Mars | 8.2 | 27.8 | 31.7 | 69.556 | 49.705 | 920–1,000 |
| Radical | 16.6 | ⟲ | | 69.547 | 49.607 | 960–1,090 |

**Table 1.** Catchment area, center coordinates, and elevation range.

The largest catchment in this study, named Radical, bounded Mars to the southeast and drained an area of 16.6 km$^2$ in 2017, spanning elevations of 960 to 1,090 m (Table 1). In 2017–2018 Radical Catchment contained two perennial supraglacial lakes (36 and 41 in Morriss et al., 2013) at elevations greater than 20 m from the terminal moulin, *Radical Moulin*.

## 2.1 Supraglacial Stream Mapping

During the 2017 and 2018 melt seasons we mapped the streams draining Mars and ArcSav Lakes from lake shoreline to terminal moulin (Fig. 1, Tables 1–2). In 2017 we conducted a roving differential GPS survey using a Trimble R7 receiver and a TRM41249.00 antenna mounted to a backpack. The receiver recorded measurements every five seconds as we traversed the edge of the supraglacial streams draining each lake. We paused every 50–100 meters along the transect to collect five or more position measurements to improve the vertical position measurements. Positions were determined with TRACK software (Herring et al., 2010) that uses carrier-phase differential processing relative to base station KAGA which has a baseline length of 28 km (Fig. 1a). After post-processing, we corrected the vertical component of our position timeseries to account for the antenna elevation above the ice surface (2.019 m). The resulting timeseries have a relative vertical error of approximately 0.04 m, which is well below the elevation change recorded during the transects (Mejia and Gulley, 2023).

In 2018, we mapped supraglacial streams using a Garmin In-Reach handheld GPS unit. We recorded positions at 100 m intervals along each stream traverse. Because Garmin In-Reach cannot accurately resolve the elevation associated with each position, we use the recorded positions to extract the elevation from the ArcticDEM strip digital elevation model (DEM). ArcticDEM utilizes submeter (0.32–0.5 m) resolution data acquired by the Maxar constellation of optical imaging satellites (i.e., WorldView 1–3, and GeoEye-1 satellites) acquired in this case between 2008–2020. The resulting DEM produces elevations referenced to the WGS84 ellipsoid with a spatial resolution of 2 m, an internal vertical accuracy of 0.2 m, and an absolute accuracy of 4 m. We find good agreement between ArcticDEM elevations and those measured by three on-ice GNSS stations in August 2018, however, ArcticDEM under-predicted elevations by 1.02 m averaged over the three stations. Throughout this manuscript we report all elevations in meters above the WGS84 ellipsoid.

## 2.2 Catchment Delineation

To delineate IDCs, we corrected automatically determined boundaries by visual inspection of remote sensing imagery. We used the ArcticDEM mosaic with a ground sample distance of 2 m (Porter et al., 2018) derived from the panchromatic bands



| | Coordinates | | Elevation | Area Drained (km$^2$) | | |
|---|---|---|---|---|---|---|
| | °N | °W | m | 2017 | 2018 | 2019 |
| ArcSav | 69.5556 | 49.7684 | 916 | 3.9 | ↘ | - |
| Mars | 69.5550 | 49.7669 | 917 | 8.2 | 3.9 | - |
| Radical | 69.5430 | 49.7007 | 961 | 16.6 | - | - |
| Phobos | 69.5592 | 49.7739 | 918 | - | 27.8 | 31.7 |

**Table 2.** Terminal moulin coordinates, elevation in meters above WGS84 ellipsoid.

of WorldView satellites in the DigitalGlobe optical imaging constellation. We project the DEMs into the WGS84/NSIDC Sea Ice Polar Stereographic North coordinate reference system (EPSG:3413), with the height above the WGS84 ellipsoid as its vertical reference.

Before catchments were delineated, we used an algorithm to identify and fill topographic sinks (Conrad et al., 2015; Wang and Liu, 2006) while preserving the downward slope of the flow path (i.e., the minimum slope gradient between cells). We use the resulting depression-free DEM to calculate supraglacial flow accumulation via the steepest descent algorithm (flow into and out of each grid element), producing a predicted channel network of supraglacial stream locations and intersections. The calculated surface routing network is used to define DEM-predicted drainage basins with catchment outlets (sinks) along the periphery of the DEM domain. We divided the large drainage basins according to the moulins identified in the field. Catchment bounds were adjusted to correct for supraglacial streams identified in WorldView-2 imagery.

## 3 Results

### 3.1 Supraglacial Stream Routing, 2017

In August 2017, Mars Lake drained through a 1.16 km long, deeply incised stream that flowed out of the lake in the opposite direction of the ice surface topography (Fig. 2). On 11 August 2017 we used a roving GPS to constrain elevations along the southern edge of the stream, starting at the lake shoreline and ending the traverse at the terminal moulin, *Mars Moulin* (Fig. 3a–b). The stream flowed west from the lake shoreline as it cut through the ridge defining the lake drainage–basin. Stream incised depth increased for 500 m as it approached a topographic high at an elevation of $926.3 \pm 0.04$ m, in agreement with a topographic high of $925.6 \pm 4.0$ m reported by ArcticDEM. This drainage divide was $11.09 \pm 0.04$ m above the shoreline of Mars Lake at the time of mapping (Figs. 3b, 4a–c). Thermal incision at the stream base created a downhill slope that allowed water to flow perpendicular to the larger-scale ice surface elevation contours containing Mars Lake. The stream curved around six snow dunes that defined the northern edge of the canyon. Snow bridges connecting to these large dunes were observed along the most canyonized reaches of the stream (Fig. 4b–c). After breaching the topographic divide, stream incised depth decreased as it approached Mars Moulin. Fresh overflow features marked the final 100 m of the stream before water flowed into the ice-rimmed Mars Moulin (Figs. 4d, 5a), located at an elevation of 917 m.





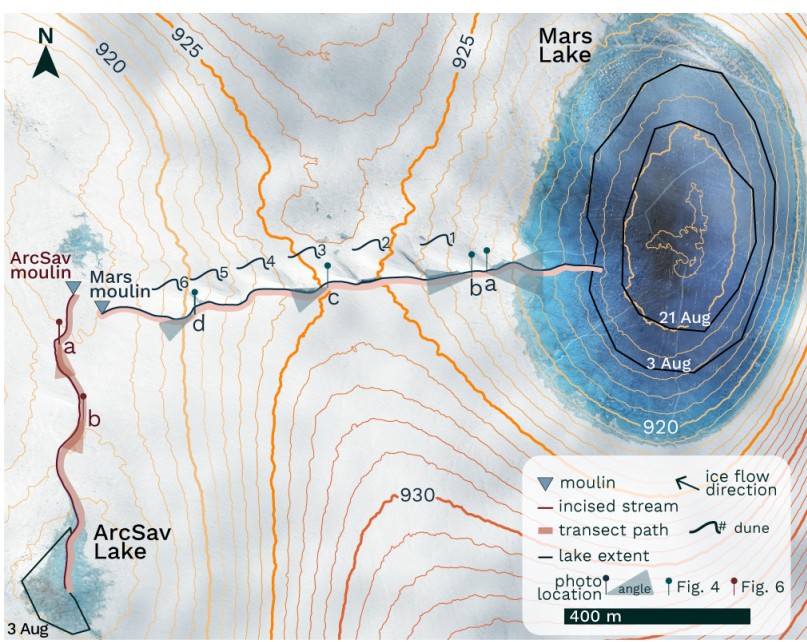

**Figure 2.** 2017 Mars and ArcSav lake drainage flow paths and mapping traverse overlaid on Worldview Imagery (©2017 Digital Globe) acquired on 3 July 2017. Pins and shaded areas mark the location and look-angle of photos in Fig. 4 (blue) and 6 (red). 2017 supraglacial lake extents are outlined in black. 1 m ArcticDEM surface elevation contours are in white.

ArcSav Lake drained through a 700 m long incised stream that flowed in the opposite direction of ice surface topography (Figs. 2, 3c–d). On 11 August 2017 we conducted a roving dGPS survey along the eastern edge of the stream draining ArcSav Lake, from lake shoreline to terminal moulin, *ArcSav Moulin*. The stream flowed north from the lake shoreline over for 180 m
at which point the streams incised depth deepened while the surrounding ice surface elevation increased by $3.86 \pm 0.06$ m over the subsequent 250 m before stabilizing at this higher elevation (Fig. 3c–d). The final 150 m of the stream was characterized by the presence of snow bridges (Fig. 6a) near where the stream incised steeply before being obscured by a thicker snow bridge that plugged the final 50 m of the stream and ArcSav moulin (Fig. 5b).

Radical Catchment, the largest catchment in this study (16.6 km$^2$), drained through a 2.4 km long supraglacial stream,
*Radical River*, into the snow-covered Radical Moulin (Figs. 1b, 5c, 7). Radical River began on the western shoreline of Lake 41 (Fig. 1b; Morriss et al., 2013), where it flowed in a direction counter to surrounding ice surface topography for 370 m, which allowed the river to breach a topographic high of 980.5 m (Fig. 7b). After breaching the topographic divide, the river flowed downslope for 1.37 km, widening as it approached a topographic low at 960 m. Radical River continued northwest until reaching the 961 m elevation contour (Fig. 7), at which point it followed the contour west, narrowing to ~1.5 m before draining
into Radical Moulin.





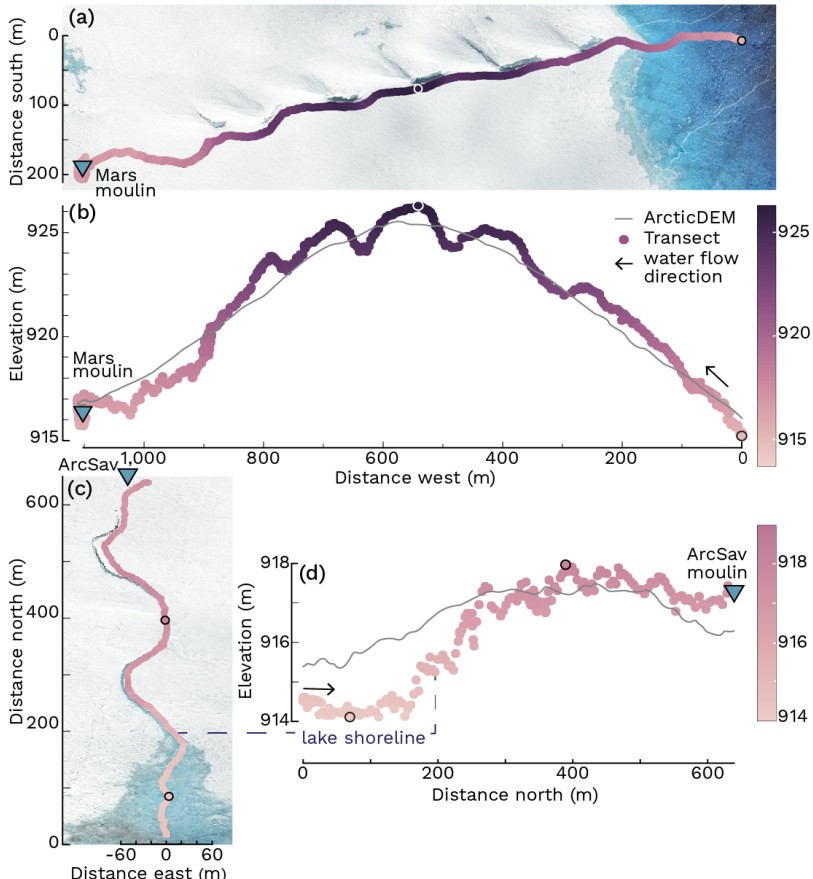

**Figure 3.** Elevations along supraglacial stream bank on 11 August 2017. Plan-view of transect for (a) Mars Lake and (c) ArcSav Lake. Elevation along stream bank in meters above the WGS84 ellipsoid reported as distance (b) west of Mars shoreline and (d) north of the ArcSav shoreline. ArcticDEM elevations along transects (gray) are included for reference. Maximum and minimum elevations along transects are outlined in a contrasting color. The colors in (b) correspond to all subplots.

## 3.2 Interannual Supraglacial Stream Routing

In 2018, Mars Lake drained through a new 1.3 km long, deeply incised stream that cut through the ridge which defined the western boundary of the lake basin, abandoning the snow-plugged channel Mars Lake drained though in 2017 (Fig. 8). On 8 August 2018 we mapped the stream draining Mars Lake and extracted elevations from ArcticDEM for positions along the

northern edge of the stream from lake shoreline to *Phobos Moulin*. Lake extent delineation from satellite imagery suggest Mars Lake began to drain in late July (Fig. 8). The new channel draining Mars Lake formed 120 m north of the snow-plugged 2017 channel. The stream incised depth increased for 370 m as it approached the topographic high at an elevation of $926.3 \pm 4$ m, which was $4.7 \pm 0.2$ m above the lake shoreline at the time of mapping (Fig. 9a–b). A comparison of ArcticDEM reported maximum elevations along the streams draining Mars Lake indicates the topographic high the lake drained through in 2018 is



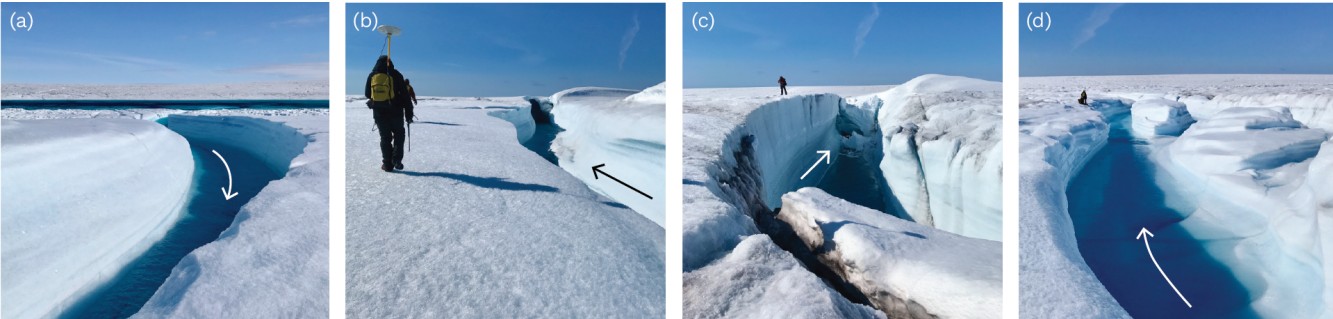

**Figure 4.** Photographs taken during the 2017 roving survey of the channel draining Mars Lake on 11 August between 13:00–14:00 LT. Arrows indicate water flow direction. (a–b) Water flow at the base of an incised channel which allowed flow in the direction opposite to large-scale topography. (c) Topographic high-point breached by the stream draining Mars Lake, a partially intact snow bridge is present. (d) Stream flow after breaching the topographic high-point, incised stream depth decreases towards Mars Moulin with overflow features apparent on the northern channel bank.

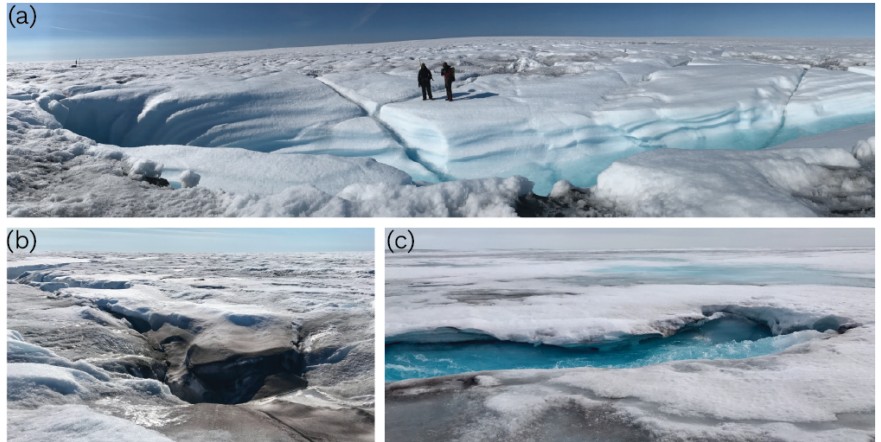

**Figure 5.** 2017 terminal moulins. Photos of (a) Mars Moulin and (b) ArcSav Moulin taken 11 Aug. 2017. (c) Radical Moulin taken 14 Aug. 2017.

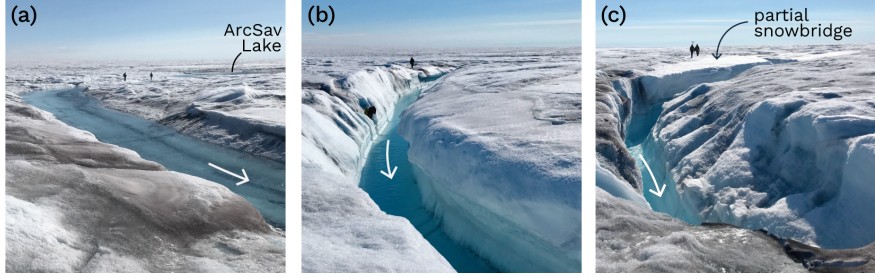

**Figure 6.** Photographs taken during the ArcSav Lake 2017 roving survey on 11 August 2017. Locations in (Fig. 2).



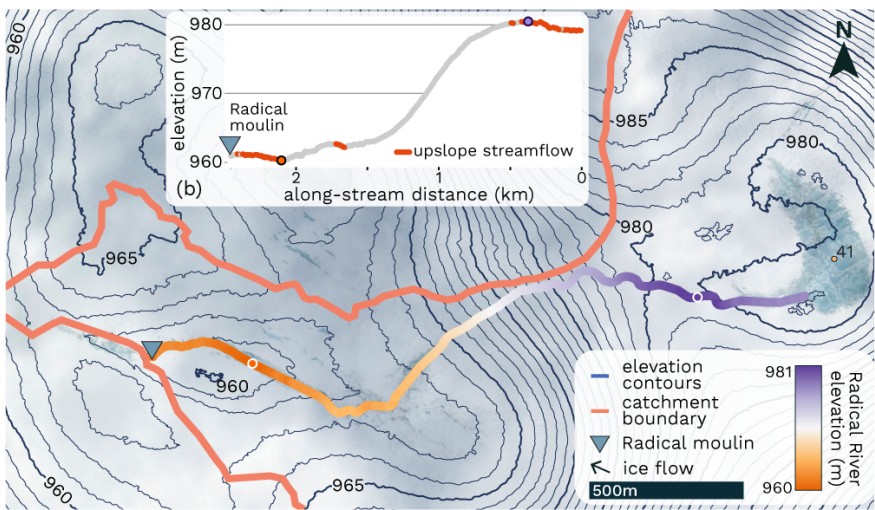

**Figure 7.** Radical River 2017 flow path. Worldview Imagery (©2017 Maxar) with ArcticDEM surface elevation contours. (b) Radical River elevation profile with uphill flow in red.

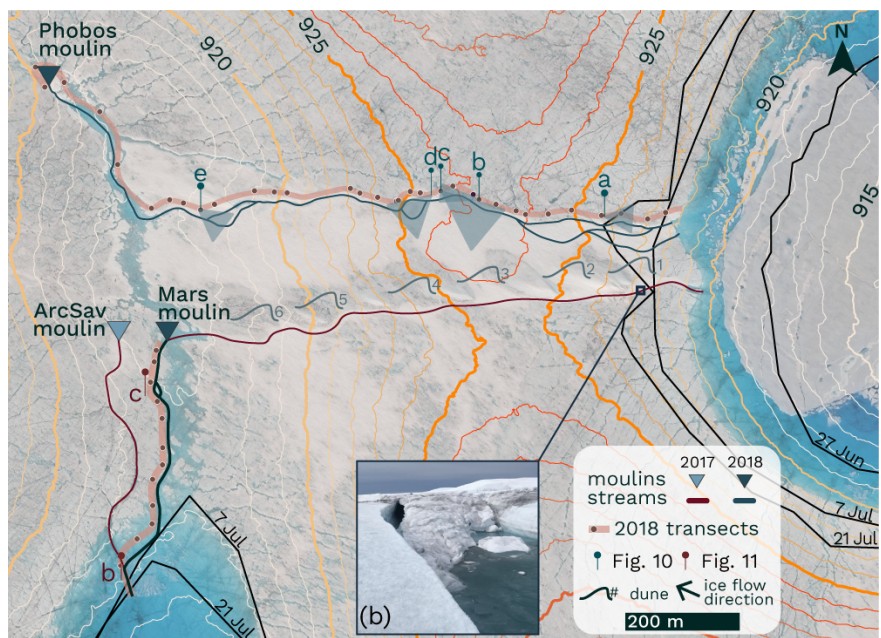

**Figure 8.** 2018 Mars and ArcSav Lake drainage flow paths. WorldView Imagery acquired 08 June 2019 (©2019 Maxar). Lake drainage paths for 2017 (red) and 2018 (blue). Symbols are as in Fig. 2. (b) Snow-filled 2017 drainage path on 6 August 2018.





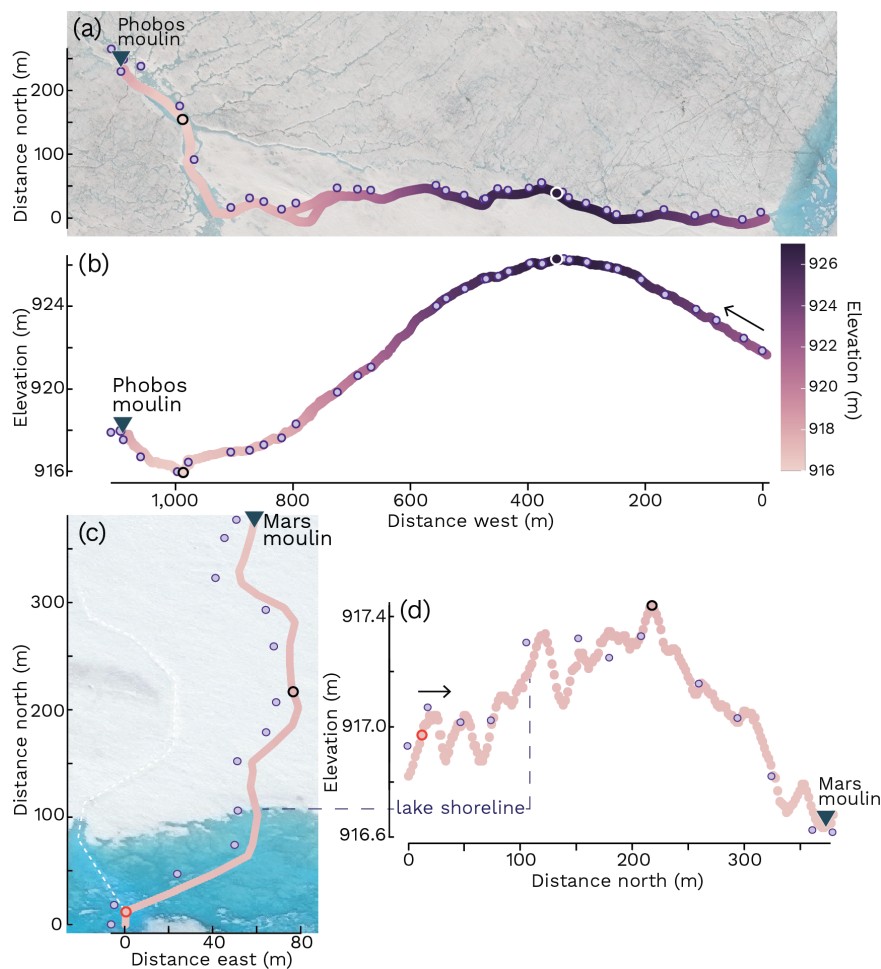

**Figure 9.** ArcticDEM extracted elevations along 2018 lake to moulin drainage paths for (a–b) Mars and (c–d) ArcSav catchments. Plan-view of transect for (a) Mars and (c) ArcSav from lake shoreline to terminal moulin overlaid on WorldView Imagery acquired on 09 June 2019 and 27 June 2018 respectively (©2019 Maxar). Stream transect elevations for (b) Mars and (d) ArcSav catchments for point measurements (purple dots) and traced supraglacial stream. Orange outline marks intersection with 2017 abandoned drainage path (white dashed line).





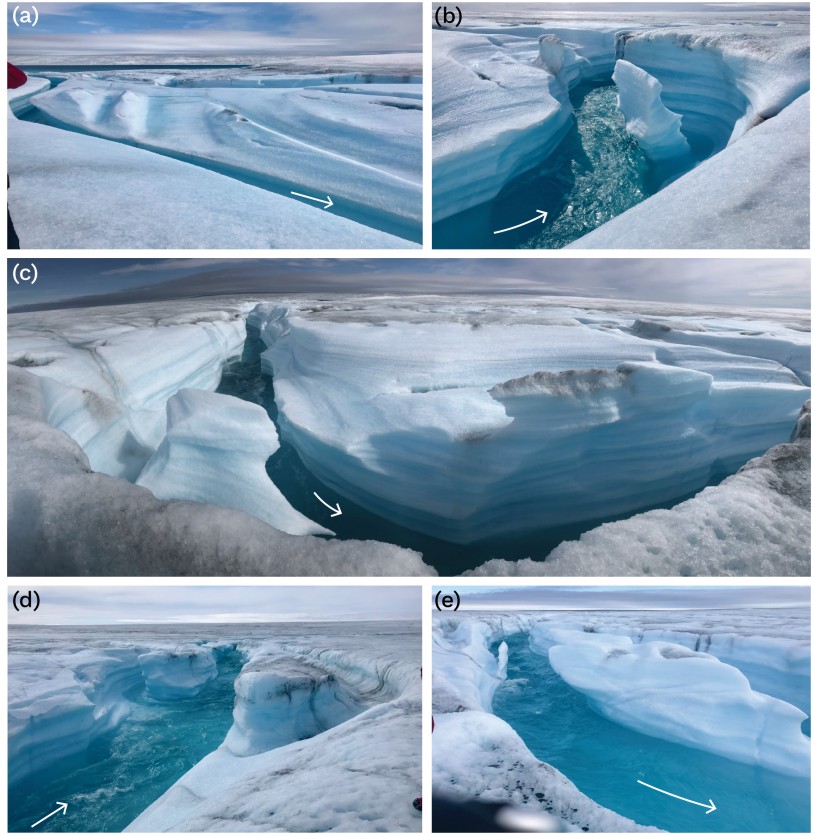

**Figure 10.** 2018 Mars Lake drainage photos. 8 Aug. 2018 15:00 LT (a) Overflow features and abandoned meandering channel formed during the initial stages of lake drainage. (b–c) Stream flow within the deeply incised channel at the the topographic high. The canyon walls show horizontal demarcations from lateral incision on the cut banks in stream meander bends. (b–d) Fluvial terraces formed during earlier phases of the lake drainage.

$0.7 \pm 0.2$ m higher in elevation than the 2017 drainage path. Stream incised depth increased with ice surface elevation, with the deepest and most canyonized part of the stream coinciding with the top of the ridge (Fig. 10b–c). After cutting through the topographic divide, the stream continued downslope for 1.4 km (Fig. 10d–e) until emptying into Phobos Moulin located 500 m northwest of Mars Moulin that drained the lake in 2017 (Fig. 11c).

In 2018 ArcSav Lake drained by incising a new, 0.5 km long channel, abandoning the snow-filled channel it drained through
in 2017 (Fig. 8). The new channel followed the 2017 drainage route for less than 100 m before diverging at nearly a right angle from the snow-filled canyon formed the previous year (Fig. 11b). The stream continued north as it incised a meandering canyon parallel to the 2017 flow path. ArcticDEM extracted elevations along the 2018 drainage path show a modest 0.4 m elevation increase over the 217 m from where the stream flowed upslope between the point where it diverged from the 2017 drainage path and where the stream breached the small topographic high of $917.3 \pm 4$ m (Fig. 9d). Ultimately, the new channel draining
ArcSav Lake reactivated and drained into Mars Moulin which was 40 m southeast of the 2017 location of ArcSav Moulin





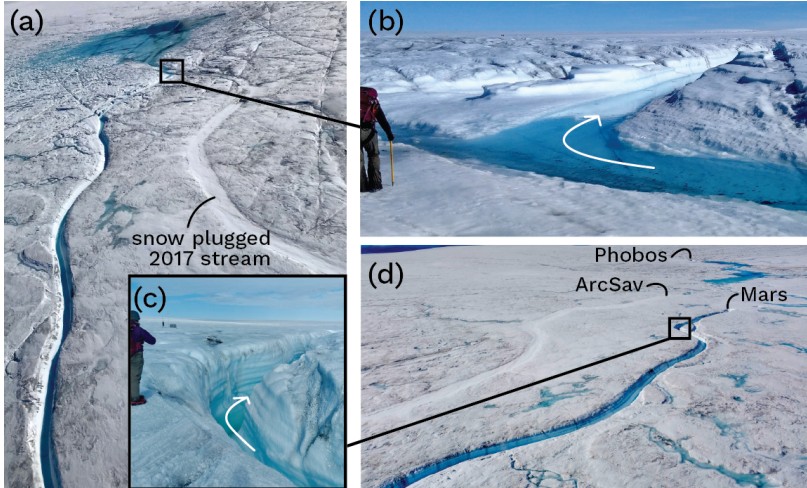

**Figure 11.** Photos of ArcSav Lake 2018. (a) ArcSav Lake, snow-plugged 2017 drainage path, and newly formed 2018 drainage path. (b) Channel intersection with snow-plugged 2017 drainage path with fresh overflow features. (c) Stream flow towards Mars Moulin at the base of a ~2 m deep channel. (d) Aerial photo of moulins.

(i.e., prior to advecting downglacier). The close proximity of ArcSav Moulin and Mars Moulin resulted in ArcSav Catchment maintaining the same area in 2017 and 2018.

 In 2018, Radical Catchment drained through the 4.8 km long Radical River that cut through the topographic high defining the western edge of the catchment and drained into Mars Lake (Fig. 12). Radical River reformed along the same path as in

155 2017, except for the final 260 m of the river which had become snow-filled and prevented the river from reactivating Radical Moulin (Fig. 13a–c). The River-snowplug interface was characterized by slush that extended several meters into the snowplug from the river edge (Fig. 13b–c). Instead of flowing towards Radical Moulin, the river widened as it flowed upslope for 630 m until it cut through the lowest point in the drainage divide separating Radical and Mars catchments (Figs. 12b–c, 13d–g). After breaching the topographic divide, Radical River flowed downslope for 2.1 km until emptying into Mars Lake. Mars Lake

160 drained into Phobos Moulin located 3.5 km northwest of Radical River which had drained the catchment in 2017. Radical River flowed against large-scale ice surface topography along 21% of the river length extending between Lake 41 and Mars Lake (delineated in Fig. 12a–b) and excluding all tributaries. Considering tributary streams totaling more than 27 km in length, streamflow against large-scale ice surface topography amounts to less than 4% of total stream length in Radical Catchment.

 During the 2019 melt season, both Radical River and the channel that drained Mars Lake reformed along the same flow

165 paths as in 2018, reactivating Phobos Moulin (Figs. 8, 12a,d). ArcSav Lake, however, drained through a new 0.9 km channel extending north of the lake shoreline, parallel to the channels draining the lake in 2017 and 2018. Instead of reactivating an existing moulin in the same area as ArcSav Moulin and Mars Moulin like in previous years, the channel continued north for 500 m and emptied into Phobos Moulin. The 500 m extension of the channel draining ArcSav Lake resulted in the merging of Mars and ArcSav catchments in 2019 and Phobos Moulin draining an additional 3.9 km$^2$ of the ice surface (Fig. 1d). As a



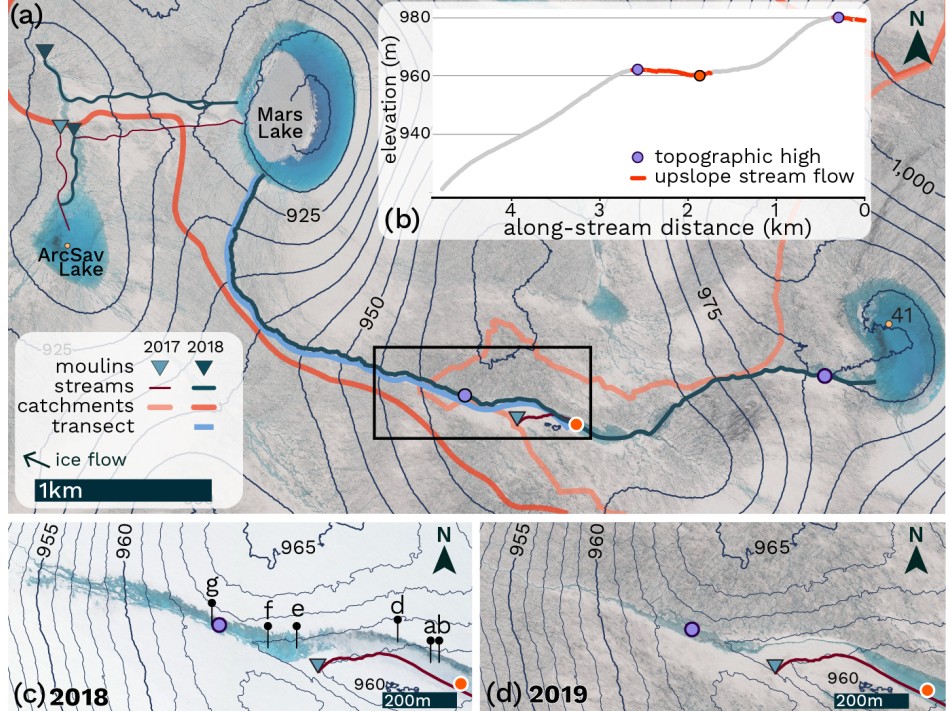

**Figure 12. Radical River breaching topographic divide**. WorldView imagery acquired on (a,d) 08 June 2019 and (c) 27 June 2018 (©2019 Maxar). (a) Overview of 2017 and 2018 stream flow routing, terminal moulins, and catchment boundaries. (b) Elevation profile with upslope stream flow (red) with circles marking the topographic high (purple) and low (orange). Radical River breaching topographic divide in (c) 2018 and (d) 2019. Pins mark photo locations shown in Figure 13.

result, all three catchments in this study merged into one large 31.7 km² catchment containing five supraglacial lakes and more than 33 km of supraglacial streams routing water into Phobos Moulin.

## 4 Discussion

IDCs are thought to be spatially fixed (Karlstrom and Yang, 2016) due to the pronounced influence of basal topography on moulin location and drainage network architecture (Crozier et al., 2018) however, our observations of the coalescence of three neighboring IDCs over three consecutive melt seasons demonstrate IDCs can vary on interannual timescales. While large-scale ice surface topography is the primary control on the flow of supraglacial streams, lakes overflowing drainage divides and subsequent thermal-fluvial erosion enables upslope stream flow relative to surrounding topography, which, even over short distances, can dramatically alter catchment-scale hydrology. Indeed, supraglacial meltwater routing generally adhered to large-scale ice surface topography with streams following the strongest downslope gradient (Crozier et al., 2018; Karlstrom and Yang, 2016). Yet, stream flow deviation from predicted paths over length scales on the order of a few hundred meters, accounting



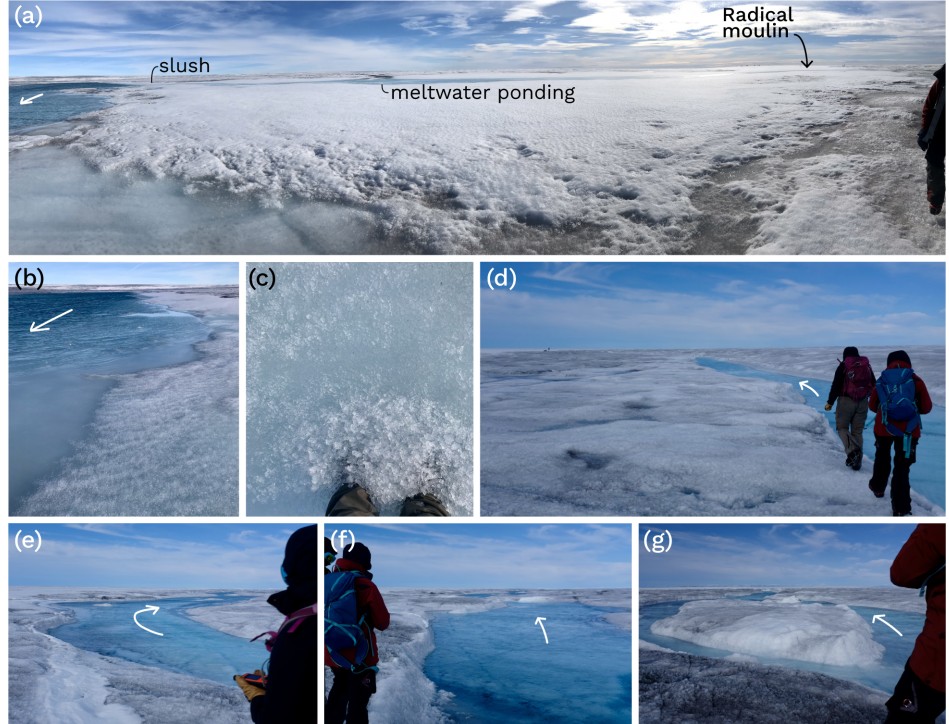

**Figure 13. Radical River mapping photos, 2018**. (a) Snow-plugged incised channel occupied during the 2017 melt season, which led to Radical Moulin approximately 200 m away (right). (b–c), Slushy transition at the stream/snowplug interface. (d–e), Radical River flowing perpendicular to ice surface elevation contours. (f) Approaching and (g) reaching the topographic high separating Radical and Mars drainage basins.

for a small fraction (< 10%) of the overall stream network length within any individual catchment, can alter catchment-scale hydrology by rerouted streamflow away from the terminal moulin of each catchment. Below we identify local controls on flow routing responsible for observed IDC interannual variability in our study area.

## 4.1 Local controls on flow routing

The slow drainage of supraglacial lakes by overtopping a spillway and draining into moulins outside of the lake basin explains the 2017–2019 drainage of ArcSav Lake. Larger July lake extents before drainage (Figs. 2, 8) coincided with the location of upslope streamflow and the low lake level recorded at the time of mapping together indicate that the drainage of ArcSav Lake began when lake water level reached the lowest point on the drainage divide to initiate a spillover event. Each year water from ArcSav Lake incised a new stream in the same location and abandoned the uplifted and snow-filled relic channel which

drained the lake the previous year. The drainage of ArcSav Lake is consistent with drainage patterns identified by Karlstrom and Yang (2016) where supraglacial streams form parallel to abandoned streams that have advected out of topographic lows.



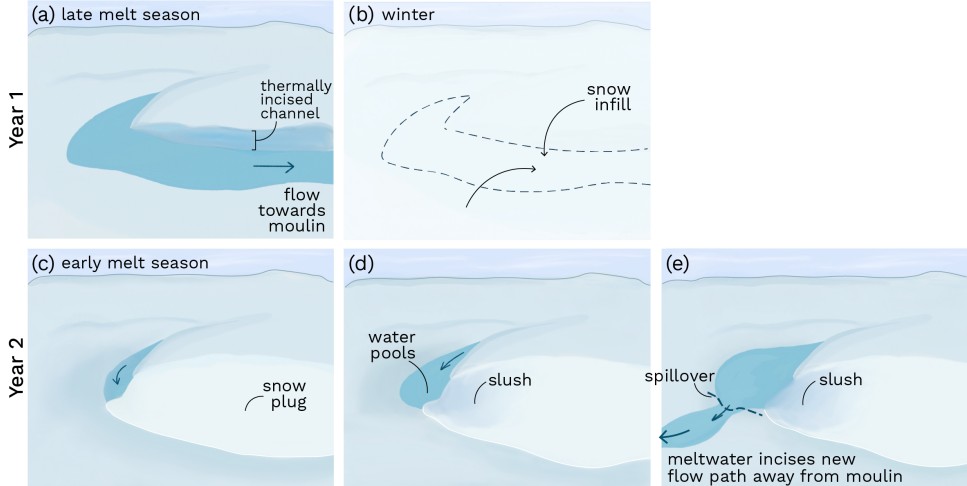

**Figure 14. Conceptual model of snow-plugged channel diverting meltwater flow**. Year 1, (a) a thermally incised supraglacial stream flows towards a moulin. (b) Snow-infill of incised stream over winter. Year 2, (c) During the following summer, supraglacial streams route meltwater to the snow-plugged incised stream formed the previous year. (d) Meltwater pools behind the snowplug while water penetrates and begins to melt the snow, creating a slushy boundary. (e) Meltwater delivery outpaces snowplug melt and pooled meltwater begins to spillover a local high-point and is rerouted away from snowplug.

The drainage location of ArcSav Catchment is therefore dependent upon the presence of a crevasse or moulin intersecting the lake drainage path, the absence of which explains the drainage of ArcSav Lake into Phobos Moulin in 2019.

The drainage of Mars Lake was more variable, accomplished by either the reactivation of relic channels or the overtopping and spillover of another point on the drainage divide. The presence of snow bridges over the most canyonized reaches of the stream draining Mars Lake (Fig. 4b–c) and gaps in snow-cover over identified in satellite imagery acquired prior to the 2017 melt season (Fig. 2), indicate Mars Lake drained by melting through the base of a preexisting snow-plugged channel that cut through the drainage divide west of the lake shoreline. After breaching the drainage divide, the stream flowed downslope until intersecting a new crevasse that hydrofractured to the bed and formed Mars Moulin. Over the subsequent winter, snow-infill of the incised channel would have reformed a snowplug meanwhile additional snow deposition overtop existing snow bridges would have further increased their height.

In 2018, instead of lake water melting through the base of the snow-plugged channel to reactive this drainage path, Mars Lake drained when lake water over-topped the second lowest point on the drainage divide and initiated a spillover event. Drainage through the second lowest point occurred because wintertime snow deposition overtop the snow-filled channel would have melted more slowly than the surrounding ice due to the higher albedo of snow, thereby increasing the height of the topographic low which originally drained the lake. The spillover of lake water through this new area of the drainage divide created a new channel that thermally incised through the topographic high and directed water away from Mars Moulin and into Phobos Moulin. Ultimately, our observations from Mars Lake demonstrate lake water can melt through snow-plugged




channels, however, when lake infill outpaces melt through the snowplug (Fig. 8b), the lake will overtop and spillover a new,
higher-elevation region on the drainage divide to form a new drainage path.

    The rerouting of Radical River, which drained Radical Catchment, can be explained by the same hydraulic processes observed within Mars Catchment, albeit at a smaller scale. In 2018, Radical River reformed over much of the same area it had occupied in the previous year, following local surface gradients until reaching a snowplug formed by the overwinter snow-infill of the most deeply incised part of the 2017 channel (Fig. 14c). Liquid water delivered to the river-snowplug intersection would
have begun to saturate and melt the snowplug, initiating some amount of porous media flow at the snowplug base (Fig. 14d). At this point, the time required to melt through and reactive the snow-plugged channel will compete against the time required to fill the surface depression to level of the next spillover point. If meltwater delivery to this depression outpaces snowplug melt, water will begin to pool behind the snowplug (Fig. 14d). Once enough water pools, it will eventually overtop and spillover the next lowest point of the surface depression (Fig. 14e), incising a new channel that could redirect streamflow away from
the snowplug. While a spillover event could occur over the snowplug itself, snow deposition overtop snowplugs, coupled with the lower ablation rates resulting from the higher albedo of snow, creates conditions that promote the tendency of snowplugs to become local high points. Where large-scale topography is shallow, as in the case of Radical River, pooling and spillover of meltwater within multiple sub-meter surface depressions could divert flow away from a snow-plugged terminal moulin and instead through a drainage divide. In this way, small scale ice surface topography can alter catchment geometry over large
scales of tens of kilometers.

    Snow infill of relic incised channels can alter IDC geometry if supraglacial streams are diverted away from the area of high extensional strain that contain new crevasses or relic moulins that previously drained the catchment. Whether water will melt through and reactivate a snow-plugged channel or be diverted away from a snowplug will be determined by the outcome of the competition between snowplug thaw and the overtopping of the local depression formed at the stream-snowplug intersection.
The rate at which pooling meltwater can melt through the snowplug will depend upon snowplug geometry, snow-water contact area, snow temperature and porosity, local surface slope at the base of the snow-plugged channel, and water temperature. The rate of depression infill will depend upon the geometry of the surface depression, the magnitude and timing of meltwater production and discharge into the surface depression. This logic could also extend to the drainage or absence of drainage of supraglacial lakes. We would ultimately expect flow diversion away from snowplugs to be favored in areas with large snow-
plugs and shallow surface slopes.

    When changes in flow routing affect lake infilling rates they can have a cascading effect wherein changes in one catchment can induce subsequent changes in flow routing within neighboring catchments. This cascading influence is illustrated by the redirection of Radical River which effectively tripled the catchment area draining into Mars Lake. This larger area draining into Mars Lake would have increased lake filling rates which likely contributed to the over-topping of Mars Lake and redirection
of flow through a new channel into Phobos Moulin in 2018. Catchment consolidation may increase lake filling rates in a way similar to high-intensity melt seasons. Although warmer years may also increase the temperature of lake water (Tedesco et al., 2012) which could hasten the ability of water to melt through snow-plugged channels. An increase in catchment area or high



intensity melting could increase lake filling rates enough to outpace snowplug melt and encourage drainage over the next lowest point on the drainage divide, potentially changing the geometry, number, and distribution of IDCs.

## 4.2 Implications


Lakes overflowing drainage divides, thermal-fluvial incision and snowplug formation should have the greatest influence on catchment-scale hydrology in high-elevation areas of the GrIS where surface slopes are shallow and moulin density is low. With increasing distance from the GrIS margin, ice surface topography shallows, the number of supraglacial lakes increases (Morriss et al., 2013), and moulin density decreases (Clason et al., 2015; Mejia et al., 2022; Yang et al., 2018). In areas where
surface slopes are strong, streams diverted away from snow-plugged channels will likely be directed back towards the area of high-extensional strain containing new crevasses or relic moulins (St Germain and Moorman, 2019). In areas with shallow surface slopes, flow diversion away from moulins should be more likely because channels would require a lesser degree of thermal incision to cut through drainage divides separating IDCs. Similar to Yang et al. (2016) who found that multi-channel supraglacial rivers are more abundant in low slope areas of the GrIS, we observed stream widening and the development of
multi-channel systems as Radical River navigated over the drainage divide separating Radical and Mars Catchments (Figs. 12c-d, 13e-g). The formation of thermally eroded anastomosing streams should assist supraglacial streams in breaching drainage divides where surface slopes are shallow.

The number of crevasses and moulins decreases with distance from the ice sheet margin, resulting in larger and fewer IDCs (Clason et al., 2015; Yang et al., 2018). In areas with high moulin density, such as on alpine glaciers or in lower-elevation
parts of the GrIS ablation area (Lampkin and Vanderberg, 2014; Yang et al., 2016), supraglacial streams will only need to flow a short distance, on the order of a few hundred meters, before reaching another crevasse or moulin in which they can drain. The proximity of other drainage locations will minimize the impact of a change in flow routing on catchment geometry or the distribution of meltwater inputs to the bed. Areas with low moulin density, such as in mid-to-higher elevation regions of the GrIS ablation area, (e.g., for elevations greater than 900 m a.s.l. in Paakitsoq), supraglacial streams will have to carry water
over long distances, on the order of several kilometers, before encountering another crevasse or moulin in which they can drain. In both cases catchments may exhibit interannual variability however, the magnitude of change in the number and geometry of IDCs and location of surface-to-bed hydraulic connections will be magnified in higher elevation areas where moulin density is low.

IDCs are important for modulating the timing and volume of meltwater delivery to moulins, with longer meltwater routing
delays for larger catchments. Heterogeneity in IDC area is reflected in the timing of meltwater delivery to GrIS moulins, where lags between the timing of peak meltwater production and peak meltwater delivery to moulins can range from 0.4 to more than 9 hours according to models (Smith et al., 2017) and in situ observations (McGrath et al., 2011; Mejia et al., 2022). The increase in IDC area with distance inland from the margin affects the timing of meltwater inputs to the bed such that peak meltwater delivery to moulins occurs earlier in the day at lower elevations, and progressively later in the day further inland
(Mejia et al., 2022), a pattern repeated in the timing of daily peak sliding speeds (Hoffman and Price, 2014). Spatial and temporal differences in peak subglacial pressurization increase the area of the bed that is resisting flow (Ryser et al., 2014), and



can dampen the effect of meltwater inputs on sliding (Crozier et al., 2018; Mejia et al., 2022). As such, changes in IDCs that reduce moulin density will further delay the timing of peak discharge into moulins while also increasing the area of the bed resisting flow. While the proportion of the bed actively connected to meltwater input drainage should scale with meltwater flux and the seasonal evolution of the subglacial drainage system (Andrews et al., 2014; Hoffman and Price, 2014), the removal of point sources of meltwater accessing the bed should result in a larger area of the bed resisting flow. Given the influence of the supraglacial drainage system on inputs to the subglacial drainage system, future work should evaluate the influence of spatiotemporal IDC variability on ice dynamics and subglacial drainage system evolution within higher-elevation regions of the GrIS ablation area.

The use of static drainage basin areas and moulin locations by surface hydrology models limits their ability to assess the impact of IDC variability on hydrodynamic coupling. Modeled moulin locations are either based on surface topography (e.g., positioned in topographic lows (Banwell et al., 2013, 2016)), or identified from satellite imagery (Koziol et al., 2017). Moulin location detection from satellite imagery is time-intensive due to the lack of automated approaches currently available, hindering the feasibility of surface hydrology models using a timeseries of moulin location to assess IDC interannual variability. We suggest an alternative approach that leverages the criteria for flow rerouting presented here with preexisting moulin locations. This approach would entail flagging moulins vulnerable to flow diversion such as those located in high-elevation areas with shallow surface slopes and where supraglacial stream incision rates are high enough to encourage snowplug formation. Flagged moulins can then be activated or deactivated in any given melt season to reflect the IDC consolidation observed in this study. With these changes preexisting models can be used to explore changes to model parameters such as lake filling rates, supraglacial meltwater routing delays, the timing of supraglacial lake drainages, and the meltwater flux into moulins. We encourage future work to explore the ice dynamic implications to IDC variability as we expect high-elevation supraglacial hydrology to become more important as the GrIS ablation area continues to expand inland in response to climatic warming (Noël et al., 2021).

## 5 Conclusions

Our observations of the coalescence of three catchments over three consecutive melt seasons from 2017–2019 demonstrates that not all IDCs are spatially fixed but can exhibit interannual variability in their number, geometry and area. Specifically, we showed that localized ice surface topography, namely snowplug formation in relic streams impounded meltwater which subsequently overflowed and incised through drainage divides, can redirect the largest supraglacial streams within a catchment away from terminal moulins and instead over drainage divides. By redirecting an IDCs largest streams, streamflow against large-scale surface topography over short distances, on the order of a few hundred meters, can impact catchment-scale hydrology ($>10$ km$^2$). We find the importance of thermal-fluvial erosion on catchment-scale flow reorganization is magnified at higher elevations distal from the GrIS margin where surface slopes are shallow and moulin density is low. Our work shows that interannual variability in IDCs is more likely to occur at these higher elevations and IDC reorganization in these areas will have a greater impact on GrIS hydrodynamic coupling. Spatiotemporal variability in IDC geometry and moulin locations



should therefore be considered in surface meltwater routing models used to explore the ice dynamic response to future melt increases.

*Data availability.* The processed data from the roving GNSS survey are archived via the Arctic Data Center and can be accessed via Mejia and Gulley (2023). Satellite imagery provided by the Polar Geospatial Center, WorldView imagery ©Maxar Inc. ArcticDEM provided by the Polar Geospatial Center and can be accessed following (Porter et al., 2018).

*Author contributions.* JM and JG planned the campaign; JG and MC acquired funding; JM, JG, CB, and CT performed the measurements; JM analyzed the data; JM wrote the manuscript; JG, CB, CT, and MC reviewed and edited the manuscript.

*Competing interests.* The authors declare that there are no competing interests or conflicts of interest.

*Acknowledgements.* This work was supported by the NSF-OPP award 1604022. We would like to thank V. Siegel and R. Knoll for their contributions to fieldwork with logistical support provided by Polar Field Services Inc. Geospatial support for this work was provided by
320 the Polar Geospatial Center under NSF-OPP awards 1043681 and 1559691. DEMs provided by the Polar Geospatial Center under NSF-OPP awards 1043681, 1559691, and 1542736. The Global GNSS Network (GGN) is operated by UNAVCO, Inc. at the direction of the Jet Propulsion Laboratory (JPL) for the National Aeronautics and Space Administration (NASA) with support from NASA under NSF Cooperative Agreement No. EAR-1261833.



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
