# Peer review of "Greenland supraglacial catchment consolidation by streams breaching drainage divides"

_EGUsphere, 2024_

## Author Comment (AC2)

**Author response to reviewer comment: Reviewer 2**

In this study, Mejia and others present field observations of supraglacial streams in the Paakitsoq region of the Greenland ice sheet that show how the drainage paths of supraglacial lakes can change considerably between different melt seasons. The study provides valuable insights into the processes by which supraglacial streams form and transport melt water from lakes to moulins, which is very relevant information for understanding and modeling the impact of surface melt on ice dynamics. The manuscript is well written but would benefit from a few clarifications especially in the methods section.

We would like to thank the reviewer for their constructive feedback our work. We believe the implementation of the reviewer's suggestions will result in the improved clarity of our manuscript, particularly in the methods sections and by incorporating all minor comments described. Below we respond to each point raised by the review with our text written in **blue** and each response will detail how the manuscript will be revised to address each concern.

Sincerely,
Jessica Mejia on behalf of all coauthors

General comments
1)  Section 2.1 describes the stream mapping associated to Mars and ArcSav lakes in 2017 and 2018, but it does not mention Radical lake, the results of which are presented in Figs. 7 and 12. Was this data acquired in the same in-situ way or from e.g. WorlView imagery? The same is true for all the streams in the 2019 melt season. Although not delineated in any map (why not?), the 2019 stream paths are described in the last paragraph of the results section. At this point, the reader has to guess that this information is based on the WorldView image e.g. in the background of Figure 12. It would be good to state this explicitly. If that is how the stream paths were determined for 2019, would it be possible to use such a remote sensing approach to map streams on a larger scale or over more melt seasons?

We will revise section 2.1 to improve the clarity of our mapping methods which, as the reviewer acknowledges, did differ between years. This section will explicitly differentiate between techniques to distinguish (1) roving differential gps surveys conducted for Mars and ArcSav catchments in 2017 (2) the stream delineation from WorldView imagery with extracted ArcticDEM elevations for Radical Catchment in 2017.  This methodology was also utilized for all streams in 2019 because we did not return to the field to make any ground-based measurements. And finally, (3) mapping supraglacial streams with a hand-held Garmin In-Reach. These positions were then used to extract ArcticDEM elevations along the stream paths for all catchments (Mars, ArcSav, and Radical) in 2018.

2) Section 2.2 is slightly short for the reader to fully understand what was done. In particular, I am wondering about two points:

a) Does the `steepest descent algorithm` (L98) refer to the Wang and Liu (2006) method? If so, I suggest citing them again, otherwise it is not clear that their method was not only used for filling the depressions. Furthermore, `the steepest descent algorithm` commonly refers to a search algorithm in optimization that has little to do with how it is used here, so I also suggest avoiding this specific term.

b) How were the DEM-predicted catchments `divided according to the moulins identified in the field` and `corrected for supraglacial streams` (L101-102)? There must be a set of rules that were followed, for instance that streams were not allowed to cross catchment boundaries, etc.? How was this done and how much ambiguity was there in this correction?

Potentially, it could also be helpful and interesting to show the difference between the DEM-predicted catchments and the corrected one. If the corrections were substantial, it would mean that topography alone was not a good predictor of flow paths in this case, which could strengthen the message of this study.

We will expand our discussion of catchment delineation in Section 2.2 to cover these points raised by the reviewer. Specifically, we will add a citation to L98, revise our phrasing for the steepest ascent algorithm, and expand upon how we refined the predicted catchment by including the rules we used to make adjustments. We will also elaborate on the phrase in L101-102 which states that the predicted catchments were "divided" according to moulins identified (explanation below). Finally, we will add a description of how much each catchment varied from the predicted boundary following manual adjustment (in the form of text and either statistics or a figure if possible).

*To quickly address the reviewers point, this is referring to the fact that our methodology to calculate the catchment requires a single outlet point which was set to the ice sheet margin, rather than the location of each individual moulin. Therefore the resulting algorithm produced one very large catchment spanning beyond our study area. We used these predicted flow paths in conjunction with observed moulin locations "divide" this large catchment into the moulin-drained supraglacial catchments discussed in the manuscript.*

3) The manuscript has many figures with partially redundant information, perhaps this could be condensed. For example, the Mars and ArcSav stream paths are depicted in Figures 1, 2, 3, 8, 9 and 12. It takes some effort from the reader to figure out the differences between those. Furthermore, it is not clear why Figure 7 has a different design than Figures 3 and 9. Does it not show the same information, just for Radical stream instead of Mars/ArcSav? Why is there no 2018 profile for Radical stream?
We will adjust figure formatting to maintain consistency between catchments. We will also explore ways to more clearly indicate the differences between figures (and rationale), potentially combining figures that show the map view of supraglacial lakes along with their stream flow data (e.g., combining Figs. 8-9) similar to what we did for Radical Catchment.

Specific comments
L35: 'through' instead of 'though'?
L117: Technically, topography does not have a direction, perhaps use slope, downward gradient or similar. (Same in L126 and maybe elsewhere.)
L127-128: `... the river flowed downslope...` could be `down the surface slope` because technically the river always flows downslope.

L148: 'stream flowed upslope' sounds like water actually flowed uphill, see L127-128; there might be other such examples that I did not point out. I understand what is meant, and it is a very minor comment, but I still think it could be more precise. Or it could be clarified once in the beginning.

L159-160: `Mars Lake drained into Phobos Moulin...` is a slightly confusing sentence. Should it be Radical Moulin instead of Radical River? And it must have drained the Radical catchment, it could be more clear to add that name again.

L162-163: Do these numbers about all tributary streams come from the DEM-predicted flow path calculations?

L186-188: The whole sentence 'Larger July lake extents before drainage coincided with... ' is unclear. How can a larger lake extent coincide with a location of `upslope streamflow`? Maybe what is meant is that it coincides with `upslope streamflow` in time? The `together indicate` does not have a proper subject in the sentence, unless e.g. `coincided` is changed to `coinciding`, if that is what is meant.

L220-222: Why is snow deposition favored on top of snowplugs? It is not just the albedo that is responsible for snowplugs becoming high points?

L222: Shallow or flat? Shallow topography usually means that the ice thickness is small. This formulation was used in other places, too.

L250: `strong` slopes seems like an unusual formulation; high slopes is more common.

L275: Hoffman and Price (2014) may not be the appropriate reference here. Without knowing the article I would think it is a study that observed the timing of daily peak sliding speeds, which is not at all the case.

L306-307: `We find that ... is magnified at higher elevations ... where surface slopes are shallow and moulin density is low.` seems too strong of a statement here since this study analyzed three catchments in one particular location. I would expect such a formulation from a study that compared the flow of many more streams on a range of elevations and surface slopes. It is something that was discussed here and is expected given the presented data, but it is not a direct finding.

We will adjust our phrasing per this suggestion

---

## Author Response (AR1)

**Author response to reviewer comment: Reviewer 1**

General comments:
The authors present an interesting and engaging manuscript combining remote sensing and field measurements to outline a interesting change to supraglacial hydrology on the Greenland Ice sheet. The quality of the writing is very high, and the paper is clear in its aims and objectives. The results are presented clearly and flow naturally into the interpretation and discussion. I feel the implications of this work could have been taken further as this is very interesting and has implications for further observations and modelling. I feel the greatest changes, albeit minor ones should be made to the study site and results sections to make them easier to follow and compare as outlined in my specific comments. For these reasons I suggest minor edits to be made to the manuscript.

We would like to thank the reviewer for their positive comments on our work and for the suggestions which have improved the quality and clarity of our manuscript. We have now incorporated the comments and suggestions raised by the reviewer in the revised manuscript. Below we have responded to each point raised, our response is in **blue**, and details how the manuscript has been updated to address each concern.

Sincerely,
Jessica Mejia on behalf of all coauthors

Specific comments
Title: The title is quite long and could be condensed to the key message.
We agree with the reviewer that the title in its current form is quite long. We will update the title to read as: *"Greenland supraglacial catchment consolidation by streams breaching drainage divides"*. This change will reduce the title's length from 26 words in the old version to 9 words.

Study Area: In this section I found myself wondering why this site was picked and how the catchments were delineated, I appreciate the method is explained in section 2.2 however a nod towards this would be appreciated.
We thank the reviewer for raising these points regarding the text in Section 2 that describes our study area. We selected this area with the goal of finding the highest-elevation catchment with a visible and accessible moulin (via helicopter reconnaissance) that the snow-line had retreated past by the time of field camp establishment on 28 July 2017. We will update the manuscript to clarify that these were the highest-elevation catchments that we could safely install a field camp and map supraglacial streams on-foot. We will also include a reference to section 2.2 on line 57 to coincide with the first mention of the area occupied by the three catchments in our study.

We added the following text to L57-60:
> These catchments were chosen because they were the highest elevation catchments with visible moulins that the seasonal snowline had retreated past at the time of camp installation on 29 July 2017. Seasonal snowline retreat was important for our field team to safely traverse the area and record the elevation transects reported in this paper. The three catchments occupied a total area of 32 km2 (see §2.2 for a description of catchment delineation).

Figure 1: I find the choice of colour for contours, particularly the 50m contour colour and stream colours very similar and slightly confusing, consider changing symbology. Stream flow direction would also be appreciated.

We thank the reviewer for directing our attention to the color choices in Figure 1. We will update the figure by choosing different colors for our elevation contour lines to make them more distinct and visible from other symbols used. We will also update the colors of our stream traces to ensure they are distinct from the color of contours and to include arrows that indicate stream flow direction for each of the terminal streams. In returning to this figure we have also noticed that we did not include an indication of ice flow direction in Figure 1b, we will update the legend to include that as well.

Tables 1 and 2: I find the arrows in the tables slightly confusing, perhaps this could be explained in the table title.

We thank the reviewer for raising this concern. We will update the text in the titles of Tables 1 to include a description of the arrows. In Table 1 the arrows indicate catchment consolidation and explain the increasing size of Mars Catchment between 2017-2019. We will remove the arrow in Table 2 and replace it with a "-" to indicate no-data for ArcSav moulin in 2018 to improve clarity.

Line 80: How much elevation change was recorded during the transects? A number here would help the justification

We agree that stating the elevation change would help with putting the error associated with our dGPS transects into context on Line 80. We recorded an elevation change of 11.09 m and 3.86 m in our transects of Mars and ArcSav Catchments, respectively. We will revise the sentence on Line 82 to include that our *"the measured elevation change, which exceeded 3 m during the transects"*.

Line 92: "by visual inspection of remote sensing imagery" is very vague, I would like some more specifics here as to how these were determined.

We agree with the reviewer that this statement was vague and will expand on our methodology for adjusting the calculated catchment bounds. Specifically, we will add a paragraph to the end of the subsection 2.2 which contains the following information:
L105-114:

*The catchment boundaries calculated from the preceding analysis were then corrected to align with supraglacial streams identified in WorldView-2 imagery with a spatial resolution of 0.5 m acquired on 03 July 2017, 27 June 2018, and 08 June 2019. Maxar (2021) reports a geolocation accuracy of approximately 5.0 m of circular error int he 90th percentile, for areas without ground control points such as on the ice sheet. WorldView scenes from each year were acquired after the melt season had begun and supraglacial streams are clearly identifiable across our study area. We compared the calculated catchment boundaries to supraglacial streams visible in WorldView imagery by inspecting the perimeter of each catchment in QGIS using a screen scale of 1:5000. We adjusted catchment boundaries by manually relocating individual points comprising the catchment polygon at a screen scale of 1:500 for areas that did not align with supraglacial stream paths identified in WorldView imagery. Adjustments made to catchment boundaries were located in areas where surface slopes are shallow and are generally less than 1°, with individual adjustments resulting in a change of ±0.1 km2 to the catchment area.*

Line 108: How much does stream depth increase?

We thank the reviewer for calling our attention to this line that does not explicitly state the stream's depth. The answer to this question is that the stream's depth increased by at least 11.09+/-0.4 m. While we report this number in the text on Line 110 we can see how by stating that the drainage divide is this height above the lake shoreline we do leave it to the reader to infer the meaning. To clarify this

sentence following the reviewer's suggestion, we will rephrase the text on Line 108 to explicitly refer to stream depth.  We have added the following to Lines 126-129:

> *"While we did not measure the stream's incised depth explicitly, we can infer that the stream's surface was at least 11.1+/-0.4 m deeper than the drainage divide and the stream base would therefore be deeper than the stream surface."*

Section 3.2: I found this section a tad hard to follow. Perhaps this could be augmented with a figure denoting a timeline for key events, perhaps combining some of the field images you have? This would help condense the information and may make it easier to follow. I found the images of varying size with little text hard to follow and this could be better communicated as this information is very valuable.

We thank the reviewer for pointing out a lack of clarity in Section 3.2. We do understand that there are many photos associated with the text and have updated the manuscript to reflect the reviewer's suggestion by consolidating Figures 2, 4, and 5 into a single figure (Figure 3 in the revised manuscript). Now instead of the reader needing to refer to several individual figures and referring back to the map with photo locations, all items are in a single figure with subplots arranged so they correspond to the image acquisition location on the main map. We have also similarly combined Figures 8, 10, and 11 (now Figure 7 in the revised manuscript) so that field photos taken along the Mars and ArcSav Lake transects are displayed in the same figure as the map-view of the study area. With this adjustment we have consolidated the information previously displayed in 6 figures into the two incorporated in the revised manuscript. The timeline issue is therefore now addressed in (1) Fig. 1 b-d showing a timeline for catchment consolidation (2) grouping the 2017 map and field photos into a single figure which keeps all features together and reduces the total number of figures that correspond with the discussion of 2017 flow routing. Finally, (3) grouping the 2018 map and field photos into a single figure, similarly keeping key elements together and reducing the total number of figures for that section.

**Author response to reviewer comment: Reviewer 2**

In this study, Mejia and others present field observations of supraglacial streams in the Paakitsoq region of the Greenland ice sheet that show how the drainage paths of supraglacial lakes can change considerably between different melt seasons. The study provides valuable insights into the processes by which supraglacial streams form and transport melt water from lakes to moulins, which is very relevant information for understanding and modeling the impact of surface melt on ice dynamics. The manuscript is well written but would benefit from a few clarifications especially in the methods section.

We would like to thank the reviewer for taking the time to review our manuscript and provide suggestions for its improvement. We have incorporated the reviewers comments in the revised manuscript which has improved the clarity of the methods section. Below we have responded to each point raised by the reviewer with our text written in **blue**.

Sincerely,
Jessica Mejia on behalf of all coauthors

General comments
1)  Section 2.1 describes the stream mapping associated to Mars and ArcSav lakes in 2017 and 2018, but it does not mention Radical lake, the results of which are presented in Figs. 7 and 12. Was this data acquired in the same in-situ way or from e.g. WorldView imagery? The same is true for all the streams in the 2019 melt season. Although not delineated in any map (why not?), the 2019 stream paths are described in the last paragraph of the results section. At this point, the reader has to guess that this information is based on the WorldView image e.g. in the background of Figure 12. It would be good to state this explicitly. If that is how the stream paths were determined for 2019, would it be possible to use such a remote sensing approach to map streams on a larger scale or over more melt seasons?

We have revised section 2.1 to reflect the suggested reorganization while incorporating an elaboration of mapping methods. Specifically, we have divided the section into three paragraphs to differentiate our mapping methods: (1) roving differential gps surveys (2) coordinate determination of supraglacial stream flow paths with hand-held inReach device with elevations extracted from ArcticDEM, and (3) supraglacial stream flow determined from satellite imagery with elevations similarly extracted from ArcticDEM. Because we use different methods of elevation determination in this study (field vs. a remote sensing product) we compare the data in Figure 3 b, d.

2) Section 2.2 is slightly short for the reader to fully understand what was done. In particular, I am wondering about two points:

a) Does the `steepest descent algorithm` (L98) refer to the Wang and Liu (2006) method? If so, I suggest citing them again, otherwise it is not clear that their method was not only used for filling the depressions. Furthermore, `the steepest descent algorithm` commonly refers to a search algorithm in optimization that has little to do with how it is used here, so I also suggest avoiding this specific term.
The reference to the steepest descent algorithm does refer to Wang and Liu (2006) we have added the suggested citation at the end of that sentence. We have also rephrased this sentence as:

"The resulting depression-free DEM was then used to calculate supraglacial flow accumulation by calculating the steepest descent for all neighboring cells across our domain, flow is then implemented as into and out of the grid elements with the steepest slope Wang and Liu, 2006), producing a predicted channel network of supraglacial stream locations and intersections."

b) How were the DEM-predicted catchments `divided according to the moulins identified in the field` and `corrected for supraglacial streams` (L101-102)? There must be a set of rules that were followed, for instance that streams were not allowed to cross catchment boundaries, etc.? How was this done and how much ambiguity was there in this correction? Potentially, it could also be helpful and interesting to show the difference between the DEM-predicted catchments and the corrected one. If the corrections were substantial, it would mean that topography alone was not a good predictor of flow paths in this case, which could strengthen the message of this study.

We have expanded our discussion of catchment delineation in Section 2.2 to cover these points raised by the reviewer. Specifically, we will add a citation to L98, revised our phrasing for the steepest ascent algorithm, and expand upon how we refined the predicted catchment by including the rules we used to make adjustments. We will also elaborate on the phrase in L101-102 which states that the predicted catchments were "divided" according to moulins identified (explanation below). Finally, we have included a description of how much each catchment varied from the predicted boundary following manual adjustment in section 2.2. We have also included a new figure (now Figure 2) that shows the calculated flow routing across our study area, generated catchment bounds, and the catchment boundaries used in this study (shown in Figure 1). We also include two insets to show examples of where the generated catchment boundaries were adjusted.

*To quickly address the reviewers point, this is referring to the fact that our methodology to calculate the catchment requires a single outlet point which was set to the ice sheet margin, rather than the location of each individual moulin. Therefore the resulting algorithm produced one very large catchment spanning beyond our study area. We used these predicted flow paths in conjunction with observed moulin locations "divide" this large catchment into the moulin-drained supraglacial catchments discussed in the manuscript.*

3) The manuscript has many figures with partially redundant information, perhaps this could be condensed. For example, the Mars and ArcSav stream paths are depicted in Figures 1, 2, 3, 8, 9 and 12. It takes some effort from the reader to figure out the differences between those.

We have taken the reviewers suggestion, and that of reviewer #1 who suggested combining several figures, and condensed some of the figures presented in the results section. Specifically, we have combined Figures 2,4,5 into a single figure, Figure 3 in the revised manuscript. Now instead of the reader needing to refer to several individual figures and referring back to the map with photo locations, all items are in a single figure with subplots arranged so they correspond to the image acquisition location on the main map. We have also similarly combined Figures 8, 10, and 11 (now Figure 7 in the revised manuscript) so that field photos taken along the Mars and ArcSav Lake transects are displayed in the same figure as the map-view of the study area. With this adjustment we have consolidated the information previously displayed in 6 figures into the two incorporated in the revised manuscript.

While these adjustments do address some concerns, there remains the suggestion of combing the map-view of Mars and ArcSav lakes with the transect elevation data as we did for Radical Catchment. While there is some redundancy because we do superimpose the background imagery on the stream's plan view observations (e.g., subplots a and c of Figures 4 and 8) we keep the background imagery so readers can clearly distinguish between the lake shoreline, terminal moulins, and contextualize the elevation profile with its location along the lake drainage path. Moreover, in our attempts to consolidate the elevation profiles for Mars and ArcSav lakes with the larger map view we were unable to find an arrangement or scale that preserved details in both the elevation plots and (Figure 9) the map (e.g., elevation contours, symbols marking field photo locations, lake shoreline, and discrete sampling locations.

 Furthermore, it is not clear why Figure 7 has a different design than Figures 3 and 9. Does it not show the same information, just for Radical stream instead of Mars/ArcSav? Why is there no 2018 profile for Radical stream?

Figure 7 had a different design than Figure 3 had for two reasons, (1) to consolidate the map view and combine it with stream profile data and (2) because we used a different methodology to determine the elevation profile of Radical River (elevations extracted from ArcticDEM) compared to those of Mars and ArcSav lakes where we performed a roving differential GPS survey and therefore could resolve elevations along those streams very precisely. We also chose this representation which consolidated the Radical River data into a single figure for 2017 because this data mainly serves as a baseline to discuss the changes in flow routing observed in 2018. Because we made this choice in our presentation of data from each location for 2017, we then use the same display format for 2017 to maintain consistency.

Why is there no 2018 profile for Radical stream?

There is a 2018 profile for Radical River, (Figure 12b (original submission) and Figure 11b in the revised submission).

Specific comments
L35: 'through' instead of 'though'? Fixed
L117: Technically, topography does not have a direction, perhaps use slope, downward gradient or similar. (Same in L126 and maybe elsewhere.) Updated all instances
L127-128: `... the river flowed downslope...` could be `down the surface slope` because technically the river always flows downslope. Updated
L148: 'stream flowed upslope' sounds like water actually flowed uphill, see L127-128; there might be other such examples that I did not point out. I understand what is meant, and it is a very minor comment, but I still think it could be more precise. Or it could be clarified once in the beginning. We added a clarification of this statement at the first occurrence on L182
L159-160: `Mars Lake drained into Phobos Moulin...` is a slightly confusing sentence. Should it be Radical Moulin instead of Radical River? And it must have drained the Radical catchment, it could be more clear to add that name again. We merged this sentence with the sentence before hand to keep Radical River as the subject which clarifies the fact that Radical River drains into Mars Lake, which then ultimately drains into Phobos moulin.
L162-163: Do these numbers about all tributary streams come from the DEM-predicted flow path calculations? Yes, we have clarified this in the text
L186-188: The whole sentence 'Larger July lake extents before drainage coincided with... ' is unclear. How can a larger lake extent coincide with a location of `upslope streamflow`? Maybe what is meant is that it coincides with `upslope streamflow` in time? The `together indicate` does not have a proper

subject in the sentence, unless e.g. `coincided` is changed to `coinciding`, if that is what is meant. ArcSav Lake's extent did coincide with the location of the sake extent, we rephrased this sentence to read "We interpret these observations as indicating that the drainage of ArcSav Lake began when lake water level reached the lowest point on the drainage divide (on the northern shoreline) and initiated a spillover event which formed ArcSav stream."

L220-222: Why is snow deposition favored on top of snowplugs? It is not just the albedo that is responsible for snowplugs becoming high points? In the case of Mars Lake, large snow dunes formed on the northern lake shoreline which likely trapped additional snow over the snow bridges than would have occurred on a flat surface

L222: Shallow or flat? Shallow topography usually means that the ice thickness is small. This formulation was used in other places, too. We rephrased to "where the slope of large-scale topography is shallow" to clarify this point.

L250: `strong` slopes seems like an unusual formulation; high slopes is more common. We rephrased to high slopes

L275: Hoffman and Price (2014) may not be the appropriate reference here. Without knowing the article I would think it is a study that observed the timing of daily peak sliding speeds, which is not at all the case. Thank you for catching this error, we have updated the citation to Hoffman et al., 2011 which is the correct citation for that sentence.

L306-307: `We find that ... is magnified at higher elevations ... where surface slopes are shallow and moulin density is low.` seems too strong of a statement here since this study analyzed three catchments in one particular location. I would expect such a formulation from a study that compared the flow of many more streams on a range of elevations and surface slopes. It is something that was discussed here and is expected given the presented data, but it is not a direct finding.
We have adjusted out phrasing, changing "is" to "can"

---

## Editor Decision (ED1)

Dear authors,

Thank you for your thorough response to reviews and revised manuscript. I am pleased to recommend publication, subject to a few minor amendments requested below. These are primarily to improve understanding through rephrasing.

Abstract: rephrase: 'Supraglacial catchments areas are thought to be controlled by the influence of basal topography on the ice surface, which produce static, topographically defined catchments areas draining into a single moulin. Our observations suggest supraglacial drainage basins are more complicated. We present evidence for lakes overtopping drainage divides, enabling connection of previously isolated adjacent basins. We document interannual variability in the size, shapeand density catchments in a 31.7 km2 area by mapping supraglacial streams within three mid-elevation catchments on the Paakitsoq Region of the Greenland Ice Sheet in 2017–2019.'

Remove 'decreasing from 3 to 1 within this area'.

L60: symbol error prior to 2.2

L58: 'The three catchments were selected because they balanced team safety and operational ease with the scientific goals: they were the highest elevation catchments below the snowline at the time of camp installation on 29 July 2017. '

L82; Split this sentence: 'The methodology applied to each catchment was based on equipment availability and ability safely access each stream. In 2019, only remote sensing tools were applied.

L85: delete 'this dGPS survey utilised' and join to previous sentence with 'using'

L134-142: really like this new discussion in the methods. You give a '±0.1 km2' adjustment – would you classify this as the uncertainty in catchment area, or can you give an estimate? Particularly in 2019 where you rely on fully remote methods, I am interested in how the Worldview + ArcticDEM errors might stack up? Could you add a sentence/short statement?

L254: remove 'yet' – you've got an 'indeed' at the start of the previous sentence already.

L256: should be 'rerouting' not rerouted?

Conclusion:

L379: remove 'number', should just read 'variability in their geometry and area'.

L383: sentence order is confusing. Maybe instead: 'When a catchment's largest streams are redirected, they may flow against the surface slop of large-scale topography for short (c. 100s m) distances, which can impact catchment scale hydrology'

L387: 'our work shows that interannual vaiability of supraglacial catchment number and geometry is possible and..'

Thank you for your patience with the review and revision process, we look forward to receiving the final version.

Liz Bagshaw, Editor